# Wearable Technologies for Pediatric Patients with Surgical Infections—More than Counting Steps?

**DOI:** 10.3390/bios12080634

**Published:** 2022-08-12

**Authors:** Ines Mack, Norman Juchler, Sofia Rey, Sven Hirsch, Bianca Hoelz, Jens Eckstein, Julia Bielicki

**Affiliations:** 1Paediatric Infectious Diseases, University of Basel Children’s Hospital, 4056 Basel, Switzerland; 2Institute of Computational Life Sciences, Zurich University of Applied Sciences, 8820 Wädenswil, Switzerland; 3CMIO Research Group, D&ICT Department, University Hospital Basel, 4031 Basel, Switzerland

**Keywords:** continuous recording, vital signs, wearable device, surgical infections, children

## Abstract

Reliable vital sign assessments are crucial for the management of patients with infectious diseases. Wearable devices enable easy and comfortable continuous monitoring across settings, especially in pediatric patients, but information about their performance in acutely unwell children is scarce. Vital signs were continuously measured with a multi-sensor wearable device (Everion^®^, Biofourmis, Zurich, Switzerland) in 21 pediatric patients during their hospitalization for appendicitis, osteomyelitis, or septic arthritis to describe acceptance and feasibility and to compare validity and reliability with conventional measurements. Using a wearable device was highly accepted and feasible for health-care workers, parents, and children. There were substantial data gaps in continuous monitoring up to 24 h. The wearable device measured heart rate and oxygen saturation reliably (mean difference, 2.5 bpm and 0.4% SpO_2_) but underestimated body temperature by 1.7 °C. Data availability was suboptimal during the study period, but a good relationship was determined between wearable device and conventional measurements for heart rate and oxygen saturation. Acceptance and feasibility were high in all study groups. We recommend that wearable devices designed for medical use in children be validated in the targeted population to assure future high-quality continuous vital sign assessments in an easy and non-burdening way.

## 1. Introduction

Infectious diseases continue to contribute to childhood mortality with around 1 million annual deaths from lower respiratory tract infections and half a million deaths from diarrheal diseases alone [1]. Vital signs are simple, cheap and very important information gathered on patients with infectious diseases; however, they are often little valued, not regularly or accurately recorded, and frequently not acted on appropriately [2]. Traditionally, the clinical diagnosis and monitoring of severe infections are based, among other factors, on discrete repeated vital sign assessments. Emerging data for patients with sepsis, however, suggest that the analysis of continuously recorded vital signs may have a higher sensitivity for the detection of new or on-going infections than evaluation of single measurements against age-adapted cutoffs [3,4]. For example, heart rate variability drops several hours before clinical symptoms of sepsis are detectable in neonates [5] and in adults after bone marrow transplantation [6,7]. During the recent pandemic, continuously assessed vital signs from wearable devices (WD) were used as an additional diagnostic tool in the detection of COVID-19 infections in adults [8,9,10]. Continuous monitoring of vital signs is associated with lower rates of cardiac or respiratory arrest in adult patients, fewer transfers to intensive care, and shorter average length of stay [11]. In pediatric oncology patients, where sepsis remains a major cause of death, feasibility of continuous recording of temperature alone has recently been studied and timely fever detection was improved with continuous monitoring over the current standard of care [12].

To date, continuous assessment can generally only be achieved using cumbersome hospital-based equipment. Medical grade WDs offer non-invasive means to measure physiological parameters and achieve early detection of acute medical problems through clinical biomarkers derived from those signals [13]. They potentially minimize interference with patient care and cost and have the potential to maximize both patient comfort and ease of measurement. This is especially relevant for children, who benefit most from non-invasive measurements; however, continuous (remote) patient monitoring with wearables has mainly been studied in adult patients [13]. Continuous vital sign monitoring using wearables in the hospital or the community setting could deliver data with sufficient accuracy and precision for their algorithmic exploitation. The ability of wearable biosensors to passively capture and track continuous health data gives promise to the field of digital health, which has recently become an area of interest for its potential to advance precision medicine [14]. Such data could support clinical decision-making for several key health problems facing children globally, including serious infections. But studies investigating the performance of wearable sensors for vital sign monitoring especially in children are scarce. This information, however, is required for the interpretation of vital signs data and quality of alarms and, therefore, essential for any potential wearable sensor before its use in clinical decision-making [11].

The primary aim of this single-center observational pilot study was to evaluate acceptance and feasibility of continuous assessment of vital signs using wearable devices in children with acute surgical infections, assuming that variability analysis has the potential to provide information about response to treatment or early signs of patient improvement or deterioration.

## 2. Materials and Methods

### 2.1. Design

This prospective, observational, single-center pilot study investigated acceptability, feasibility, validity, and quality of the data collected from December 2018 to July 2019 by continuous telemonitoring on a pediatric surgical ward of the University Children’s Hospital Basel, Switzerland.

### 2.2. Participants

A total of 21 children (15 male, 6 female) with a mean age of 9.1 (range 4–17) years were enrolled and wore the device (for up to 5 days). Participants were inpatients requiring intravenous antibiotic therapy for either perforated appendicitis (*n* = 13), osteomyelitis (*n* = 5) or septic arthritis (*n* = 3). No values could be collected for one patient (ID 18) due to a problem in the data transfer. Patients, as age appropriate, or their legal guardians gave written informed consent for taking part after research assistants reviewed instructions on use and care for the devices with them. No financial or other compensation was given to patients or their parents.

Participants were asked to wear the WD on a strap located on the right upper arm during day- and nighttime. The device was to be removed during imaging, medical interventions, or personal hygiene (device is waterproof, but contact with soap, detergent, chlorinated water, salt water, and lotion should be avoided).

Exclusion criteria were skin diseases in the area where the sensor was to be worn, allergies to plastic and/or latex, expected duration of hospitalization <24 h after informed consent, upper limb impairment or disability affecting the quality of measurements such as wounds, intravenous access, tattoos, and patients who were unable or unwilling to follow study-specific instructions and examinations. Once per day, a member of the study team visited the patient and replaced the sensor with a charged device.

### 2.3. Data Collection

#### 2.3.1. Wearable Device Continuous Measurement

Continuous monitoring of the following selected parameters was carried out with a WD (Everion^®^ 3.06 consumer release, Zurich, Switzerland) by Biovotion (now Biofourmis, [15]): heart rate (HR; measurable range: 30–240 beats per minute), heart rate variability (HRV; 0–255 ms), blood oxygenation (SpO_2_; 65–100% at rest, 80–100% under motion) and skin temperature (T; 0–60 °C). These vital signs were measured with a sampling frequency of 1 Hz using photoplethysmography or infrared sensor for temperature.

The WD offers some characteristics that we judged to be important for continuous measurements in a pediatric cohort: light weight (approx. 40 g), small size (approx. 70 × 50 × 12 mm), no buttons or cables, thus offering maximal mobility to the patients, and application by elastic bands on the upper arm that do not hamper the children during their daily activities.

The patient’s vital signs data were automatically captured from the WD via a Bluetooth/Wi-Fi gateway device (Raspberry Pi 3 Model B, Raspberry Pi Foundation, Cambridge, UK) installed near the patient’s bed. As soon as the device was within reach of the gateway, a connection was established to collect and transfer the latest data to the Device Management Server for further processing.

For our data analysis, raw data were used and analyzed retrospectively. Only the principal investigator was able to access the vital sign data during the study, but not the participants or the treatment team. The device assigns a quality score (range 0–100) to some of the assessed vital signs. The data used for our calculations were filtered, retaining only measurements with an HR or SpO_2_ data quality above 50% and values larger than 0. As reported by Biofourmis [15], only measurements with a quality above 50% can be considered trustworthy. For the validation of the wearable data against conventional measurements, the wearable data were aggregated. For each conventional measurement, we computed the arithmetic mean of the corresponding filtered vital sign data in a time window ± 15 min around the conventional discontinuous measurement.

#### 2.3.2. Conventional Clinical Measurements

The conventional clinical measurements were collected according to the institutional protocols by trained clinical nursing staff at the bedside and retrieved from patient charts by the study team. Heart rate and SpO_2_ were measured intermittently using the Masimo Root^®^ Platform (Masimo International, Neuchâtel, Switzerland), and body temperature was assessed by a tympanic thermometer.

#### 2.3.3. Outcomes

The primary objectives of this pilot study were to assess (1) acceptance and (2) feasibility of the routine use of a WD for continuous vital sign monitoring in pediatric inpatients with defined surgical infections.

Secondary objectives were: (3) to describe variability and agreement of WD measured vital signs compared with conventional measurements; (4) to describe availability of vital signs during inpatient treatment captured using WDs or conventional measurements; (5) to describe reliability of vital signs data measured by WDs or conventional measurements; and (6) to analyze heart rate variability as a potential biomarker providing information about response to treatment or early signs of patient improvement or deterioration.

#### 2.3.4. Assessment of Acceptance and Feasibility

The acceptance of using the WD was measured using a seven-item questionnaire that assessed the individually perceived mobility reduction, wearing comfort, and difficulties in handling. Responses were measured on a 6-point Likert-type scale, scored 1–6, and ranged from “strongly disagree” to “strongly agree” (Table 1). For example, participants were asked to rate how much they agreed with items such as “The device was comfortable”. There were three versions of the questionnaire, with adapted language for the subgroups of adolescents (14–16 years), school-age children (11–13 years) or parents (who were asked to complete a questionnaire for their 4–10-year-old children).

#### 2.3.5. Statistical Analysis

Because this was a pilot study, we did not conduct formal power calculations for sample size estimations. The required sample size to detect a reported acceptance of the wearable device of 80% (alpha 0.05, power 0.8) assuming a 50% acceptance under the null hypothesis (i.e., indifference) was 14 patients.

We used descriptive statistics to characterize the study sample, survey responses for acceptance and feasibility (Tables 1 and 2), and variability of measured vital signs (Appendix A). In Appendix A and to display (good) data availability in Figure 1, the following quality filters were used: skin temperature (Temp > 0, HR > 0, HRQ ≥ 50), heart rate (HR > 0, HRQ ≥ 50), and oxygen saturation (HR > 0, SpO2 > 0, SpO2Q ≥ 50).

Reliability was evaluated using Bland–Altman plots, mean differences, and 95% limits of agreement (LoA) per vital sign measured by the WD compared with conventional measurements. The corresponding means of vital sign aggregates (recorded within 15 min) were calculated to account for imprecision of reported time points.

Version 3.8.0 of the Python software was used for analysis and graphical presentation (https://github.com/hirsch-lab/mhealth/releases/tag/study_ukbb_v1.0, accessed on 16 June 2022).

## 3. Results

### 3.1. Acceptance

Patients and health-care workers reported being satisfied with wearing the WD. All study groups agreed that wearing the device was considered not to be stressful or restrictive (reflected by an overall item score ≤2.5, Table 1).

**Table 1 biosensors-12-00634-t001:** Health-care workers’, parents’, and children’s’ reported acceptability of the Everion^®^ WD.

Questionnaire Items ^1^ (Likert Scale)	HCW: Doctors (n = 21)	HCW: Nurses (n = 21)	Parents (n = 16)	Adolescents 14–16 y (n = 4)	Children 11–13 y (n = 3)
How stressful did you find carrying the sensor (for your child/patient)	1.43 ± 0.21	1.33 ± 0.20	1.69 ± 0.38	1.75 ± 0.42	1.33 ± 0.53
How restrictive did you find wearing the sensor (for your child/patient)	1.29 ± 0.19	1.38 ± 0.21	1.56 ± 0.42	1.00 ± 0	1.67 ± 0.53
How was the wearing comfort during day-time (for your child/patient)	1.57 ± 0.25	1.67 ± 0.24	1.69 ± 0.38	1.50 ± 0.49	1.33 ± 0.53
How was the wearing comfort during night-time (for your child/patient)	2.00 ± 0.37	1.95 ± 0.31	2.13 ± 0.45	1.25 ± 0.42	2.67 ± 1.07
How was the wearing comfort during 24 h (for your child/patient)	1.91 ± 0.29	1.84 ± 0.30	2.06 ± 0.44	1.50 ± 0.49	1.50 ± 0.49
How did you (your child/patient) like the wearing position on the upper arm	1.57 ± 0.25	1.52 ± 0.25	1.63 ± 0.29	1.75 ± 0.42	1.67 ± 0.53
How was it to attach the sensor without help (for your child/patient)	1.85 ± 0.37	1.67 ± 0.44	2.14 ± 0.62	1.75 ± 0.81	1.33 ± 0.53

^1^ Item scores are presented as mean (± standard deviation). Duplicate answer possible. Abbreviations: HCW, Health-care workers.

Feasibility was assessed through seven open-ended questions such as “if you removed the sensor, tell us why” (Table 2). All data were collected on paper and transferred to the REDCap electronic data capture tool at the study site [16].

The wearing comfort during daytime was rated “very comfortable” (adolescents and children) or at least “comfortable” (health-care workers, parents). The only item with an item score above 2.5 was the assessment of the wearing comfort during night-time (rated “rather not comfortable” by children 11–13 years). The wearing position on the upper arm was reported being “comfortable” by all study groups. The WD attachment without help was rated to be “very easy” by the children (11–13 years), “easy” by adolescents and health-care workers and “rather easy” by the parents of children younger than 11 years of age.

Most doctors (19/21, 90%), nurses (15/21, 71%), parents (14/16, 88%), adolescents (3/4, 75%) and children (3/3, 100%) would recommend the WD to family or friends. The same was true for the question if parents (12/16, 75%), adolescents (3/4, 75%) or children (3/3, 100%) would again participate in the study. 

### 3.2. Feasibility

Most of the participants provided responses to the seven open-ended feasibility questions. The most common themes regarding what participants liked about the WD were the improved mobility compared to regular (wired) sensors, less light and noise emission, and the continuous measurement of vital signs (Table 2).

**Table 2 biosensors-12-00634-t002:** Themes endorsed by health-care workers, parents, and children on open-ended feasibility questions.

Questionnaire Items ^1^ (Open-Ended Questions)	HCW: Doctors (n = 21)	HCW: Nurses (n = 21)	Parents (n = 16)	Adolescents 14–16 y (n = 4)	Children 11–13 y (n = 3)
Reasons why you removed the sensor	n.d.	n.d.	Not removed (2), Shower (4), Disturbing sleep (4), Disturbing daytime activities (1), Medical intervention (1), Child irritable (1), Non-usable disclosures (1)	Shower (4), Battery change (1)	Shower (1)
Suggestions for improvement of the sensor	Nothing (6), Smaller (4), Integration in clothes (1), Waterproof (2), Non- usable disclosures (1)	Nothing (3), Waterproof (2), Smaller (5), Color (1)	Nothing (5), Design/color (2), Smaller (4), Attaching without help too difficult (1), Non-usable disclosures (1)	Nothing (1), Waterproof (1)	Nothing (1), Smaller (1)
Advantages compared to conventional measurement	None (1), Mobility (13), Continuous measurement (1), Non-usable disclosures (2)	Mobility (13), Continuous measurement (1), No emission of light during night-time (1), Stability (1), Non-usable disclosures (1)	Mobility (13), Comfort (1), No disturbing noise (1), Continuous measurement (1)	Mobility (2), Comfort (1), Yes (1)	n.d.
Disadvantages compared to conventional measurement	None (8), Error prone (1), Battery (1), Less clinical assessments (1), Size of sensor unsuitable for small children (1), Central monitoring (1), Patient cooperation necessary (1), Skin irritation (1), Prone to theft (1), Non-usable disclosures (1)	None (9), Measured vital signs not visible for nurses (1), No experience (1), Skin irritation (1)	None (8), Data privacy (1), “Emissions” (1), Battery (1), Measured vital signs not visible for parents (1)	None (3)	n.d.
Better or worse compared to conventional measurement	n.d.	n.d.	n.d.	n.d.	Better (3)

^1^ Duplicate answer possible. Abbreviations: HCW, Health-care workers. N.d., not done.

Children and adolescents only removed the WD for showering or medical interventions. Parents of younger children (i.e., <11 years of age) reported taking the device off if they felt the child’s sleep or daytime activities to be disturbed (n = 4 and n = 1, respectively) or if parents had the impression that the child generally did not feel well due to the underlying disease.

The most reported challenges of wearing the WD and suggestions to improve it included the device’s size (suggestion to make it smaller) and water resistance (suggestion to make it waterproof).

### 3.3. Variability and Vital Sign Agreement

Appendix A summarizes mean, standard deviation, difference (absolute and relative) and counts of each vital sign measured by the WD or conventionally.

For HR, the mean values measured by the WD and by nurses were comparable (mean difference, 2.5 bpm) as well as for SpO_2_ (mean difference, 0.4%). The WD showed consistently lower mean temperatures of 1.7 °C compared with nurse measurements, likely explained by the fact that the device measures skin temperature with a lower normal range compared to conventionally measured tympanic temperature, which is closer to core body temperature [17].

### 3.4. Data Availability

Figure 1 shows the wearable recording of data in different quality. Additionally, discrete conventionally assessed vital signs (by the medical team) are displayed by vertical bars. This information is displayed for all 20 study patients (belonging to different age groups) and during daytime versus nighttime.

Several time periods of up to 24 h without any vital sign data available were noted in some patients. These episodes cannot be fully explained by the fact that the sensors had been removed from the patients (e.g., for personal hygiene or medical interventions). Compared to the number of conventional measurements, wearables data provides more information, but missing data periods are relevant.

**Figure 1 biosensors-12-00634-f001:**
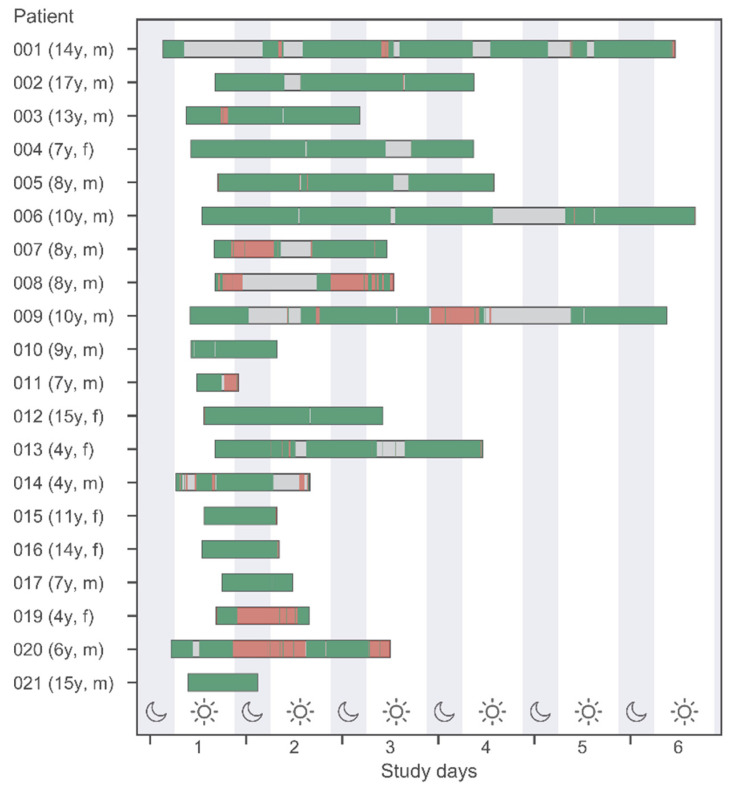
Data availability of measured vital signs by the Everion^®^ WD and conventional measurements at night- and daytime (22 h-06 h-22 h). Green and red bands indicate good and bad quality data according to quality filters described in Section 2.3.5 “Statistical Analysis”. Gray bands indicate no data available.

### 3.5. Data Reliability

Bland–Altman (BA) plots in Figure 2 show the paired WD and conventional measurements (aggregated means covering ± 15 min) with the mean difference and limits of agreement (LoA, mean difference ± 1.96 standard deviation of the difference), for heart rate and temperature (not enough data points were available for SpO2).

For HR, LoA ranged from ± 55 bpm, with a mean difference of 5.5 bpm. The WD consistently underestimated temperature with a mean difference of 2.3 °C on average, with LoA of ± 4.5 °C.

### 3.6. Exploration of Heart Rate Variability as Potential Biomarker

Two episodes of clinical deterioration with necessary surgical intervention were recorded in two different patients (No. 2, 14) during the study period. Visual exploration of the WD-recorded vital signs within 48 h preceding these episodes were hampered by missing data and did not allow us to detect a potential pattern.

## 4. Discussion

Small wearable, wireless devices that can be worn on the body are beginning to transform health care, making medicine more predictive and personalized. In this study, vital signs measurements by the Everion^®^ WD were assessed for acceptance and feasibility in a pediatric cohort, and data were compared with conventional clinical staff measurements for variability, availability, and reliability.

Results indicate that the use of a wearable device is highly accepted and feasible for health-care workers, adults, adolescents, and children. Parents and children, on average, adhered to wearing the device for most desired monitoring days, indicating the potential for the use of monitoring devices for continuous vital signs measurement in or outside the hospital. Although participants found wearing the device to be both acceptable and feasible, the results of this study show some ways the device needs to be improved for future medical use in children.

Heart rate and oxygen saturation measured by the WD were in strong agreement with current nurse measurements, whereas body temperature was consistently underestimated by the WD due to different measurement techniques (skin versus tympanic temperature). Reliability analyses showed that WD measurements of heart rate were close to nurse measurements in mean difference. Similar results for heart rate and temperature have been found in earlier studies on the Everion^®^ WD in adult patients [11].

The Everion^®^ WD is one of the few commercially available wearable sensors that measures SpO_2_, but little is known about the accuracy of measuring SpO_2_ by wearable sensors. Our results showed a mean underestimation for SpO_2_ measured by the WD of 0.4%. However, this difference needs to be interpreted carefully due to generally low availability of SpO_2_ data in our study. This might be related to patient movement (as described, for example, by Weenk et al., for adult patients [18]), likely because the WD calculates an accuracy metric per vital sign that prevents data with accuracy < 50% from being stored. The placement site (upper arm) of the WD may have played an important role as well, because this is a nontraditional and uncommon site to measure PPG signals [19].

The consistent 1.7 °C underestimation of temperature by the WD can be explained by the differences in measurement technique. Nurses measured tympanic temperature, and the WD has a thermistor to measure skin temperature at the upper arm. The advantage of measuring skin temperature especially in children is the ease of access for the thermometer. However, marked temperature gradients may develop due to environmental influences, such as the position of the arm above or under the blanket or coverage by cloth. Though the skin temperature is heterogeneous and vulnerable to ambient environment, it correlates to the core temperature as the main path to conduct heat exchange with the environment [20]. Hence, it could still be a plausible index reflecting the change in core temperature according to the site of measurement and the level of activity.

The penetration of wearable sensors into the healthcare market has been relatively slow, despite the rapid development of devices in the lifestyle and fitness markets. Even when a device has passed through the stages for medical approval, as the WD had in our study, there are still limitations that need to be considered before these devices can be used in patient care. The inability to form stable, intimate skin contact remains a fundamental constraint in their measurement capabilities, especially for sensors that are not patched to the skin and where the attachment (such as an elastic band for the Everion^®^) needs to be adapted to various sizes according to the child’s age and weight. For applications in fitness and wellness, where regulatory oversight is minimal, these restrictions do not impede the public adoption of such consumer health wearables for simple measurements of basic parameters, such as heart rate. But once a device is developed or approved for medical use, these restrictions must be overcome, and data scientists, engineers and medical specialists must carefully balance function against fashion (form). Moreover, the ability to continuously monitor parameters associated with an individual’s health state results in a high volume of data, which presents both challenges and opportunities for data analysis.

The WD was highly accepted in our study population by the patients, parents, and health-care workers, but some of the predefined study objectives were not met. The small number of patients precluded an analysis of patient factors influencing the primary outcome and allowed no exploration for specific patterns preceding episodes of clinical improvement or deterioration. The planned exploration of heart rate variability as potential biomarker was not feasible because there were not enough data. Devices that are designed for medical use must be evaluated in the targeted population, i.e., in our case in a cohort of hospitalized (sick) children. The WD was a priori intended for use in adults, which might explain the lower quality of assessed data in our pediatric cohort.

The strengths of the current study include the use of a commercially available WD that can be accessed by the general population to continuously track their vital signs. This study also included the assessment of various groups, i.e., health-care workers, parents, and an unbiased sample of adolescents and children, who could all potentially benefit from future continuous vital signs assessment. Exploring these study groups’ perceptions of the device’s acceptability and feasibility separately helped to elucidate the most important challenges they might experience when using such devices.

The wearable sensors of the current generation can track biophysical signals, such as cardiac rhythms, breathing, temperature and motion. More advanced systems are emerging that can measure certain biomarkers (such as glucose [21]) as well as actions such as swallowing and speech. The development of technologies that overcome limitations associated with loose skin contact and that incorporate advanced biochemical/biophysical sensing have the potential to transform consumer wearables from recreational novelties into body-worn, clinical-grade physiological measurement tools that yield physician-actionable information. Such systems can be enhanced using artificial intelligence to monitor vital signs, detect abnormalities and track treatments [22]. Hardware is largely covered by existing frameworks; algorithms are not. In this already current future, wearable systems will collect ever-increasing amounts of data, which will need to be transmitted via an appropriate infrastructure to databases and processing units that aggregate and analyze patient data from multiple sources. Data security must be a top priority, particularly for patient-identifiable information [23]. For hospitalized patients, their healthcare provider must act as controller and protector of their data, preventing commercial exploitation of medical data without approval and informed consent. Equally, commercial interests should not determine who can and cannot access this technology. Given the poor track record of private companies in protecting consumer privacy, leadership at both the national and international level is needed. Technical progress will require close collaborations between materials and device engineers, data scientists and medical professionals. Users and caregivers need to be more closely involved. It remains to be seen how these sensor systems will be paid for, and how medical staff will be reimbursed for interpreting and acting on the data [24]. Despite these challenges, wearable sensors have the potential to transform nearly every aspect of medicine, and wireless health monitoring could also revolutionize health care in countries where hospital infrastructure is lacking and the ratio of staff to children cared for by HCW with variable training and competencies is low [25].

## 5. Conclusions

Recent developments in non-invasive vitals monitoring with small, wearable devices open the opportunity to record high-quality vital parameters over many hours in an easy and non-burdening way. The present study demonstrated that this form of vital sign measurement is well accepted in the hospital setting from pediatric patients, parents, and the medical team. However, intimate skin contact remains a fundamental constraint in their measurement capabilities. The results of this study will influence the design of future studies on wearable devices, including those intended to identify patterns that predict patient improvement or deterioration during infection or treatment.

## Figures and Tables

**Figure 2 biosensors-12-00634-f002:**
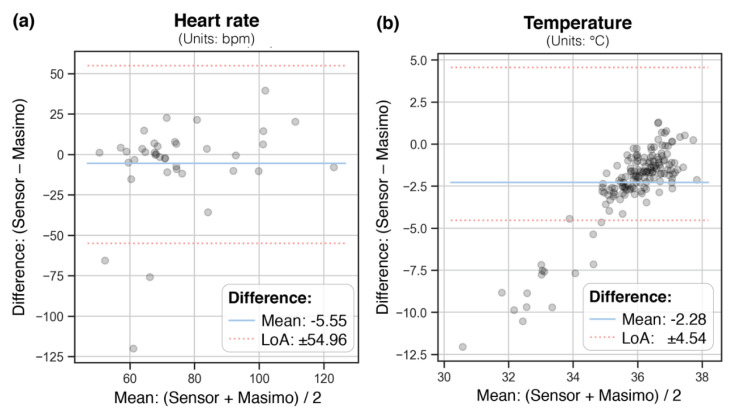
Bland–Altman plots for heart rate (**a**) and body temperature (**b**). The dots represent the measurements by the Everion**^®^** WD (Sensor) vs. conventional measurements (Masimo). Horizontal lines indicate mean difference (center line) and the upper and lower limit of agreement (mean difference ± 1.96 * standard deviation). Abbreviations: bpm, beats per minute; LoA, limits of agreement.

## Data Availability

Not applicable.

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
