# Peer review of "Wearable Technologies for Pediatric Patients with Surgical Infections—More than Counting Steps?"

_biosensors, 2022, doi:10.3390/bios12080634_

Round 1

Reviewer 1 Report

1) It is desirable to extend the introduction part and add more literature review related to the topic in other chapters, too.

2) It would be desirable if the authors provide technical description of the used sensors.

3) Consider checking the Fig. 2 and please try to increase the font size to make it more readable.

Reviewer 2 Report

In this manuscript, to describe the acceptability and feasibility of WD for continuous vital sign monitoring and to compare the data validity and reliability with traditional measurement methods, the authors used a multi-sensor wearable device (Everion®) to continuously measure vital sign data in 21 hospitalized pediatric patients with appendicitis, osteomyelitis, or septic arthritis.

Specific comments and suggestions are below:

1.     Details of the error in the paper

①Whether the title "1.Introduction" "4. Discussion""5. Conclusions"should be indented

②The format of sections 3, 4 and 5 in Section 2.3 is incorrect and should be changed to 2.3.3, 2.3.4 and 2.3.5, and the font is inconsistent.

③Figure1 and Figure2 should have the same indentation amount?

2.     Do the results obtained from Figure2's Bland-Altman (BA) diagram illustrate the reliability of the data, on what basis, and whether the conclusions are clear? The author is requested to elaborate..

3.     Is the explanation in section 3.6 sufficient to verify the conclusion that "heart rate variability is not feasible as a potential biomarker"? With the continuous improvement of technology, the improvement of data loss and the collection of more patients' vital sign data, can the above conclusion be denied?

4.     If skin contact remains a fundamental limitation of WD's ability to measure, should similar individual differences be considered in the testing process, even though children's weight or body size does not vary much?

Reviewer 3 Report

This compare the wearable devices with the coventional equipment for  diagnosing paediatric patients with surgical infections, demonstrates the feasibility and importance of biomedical wearable devices.  Furthermore, the shortcomings of the current wearable devices were also discussed, which reveals the developments challenge of the wearable biomedical devices. I think this work is important for the development of wearable biomedical devices. Before its publication, I suggest the authors to do the following revises.

1. It is seggested to provide the photo of the wearable devices used in this investigation.

2. The literatures should keep a uniform format.

Reviewer 4 Report

Congratulations for your research and publication. Very useful!

Round 2

Reviewer 2 Report

all set!